



# Major flood dominates 14 year sediment and nutrient budgets for two subtropical reservoirs

Katherine R. O'Brien[1], Tony R. Weber[2], Catherine Leigh[3,4] and Michele A. Burford[3]

[1]School of Chemical Engineering, The University of Queensland, St Lucia, Queensland 4075, Australia
[2] Integrated Catchment Assessment and Management Unit, ANU College of Medicine, Biology and Environment, Australian National University, Canberra, ACT 0200 Australia
[3] Australian Rivers Institute, Griffith University, Nathan, Queensland 4111, Australia
[4]IRSTEA, UR-MALY, 5 rue de la Doua, CS70077 69626 VILLEURBANNE Cedex, France

*Correspondence to*: Katherine R. O'Brien (k.obrien@uq.edu.au)

**Abstract.** Accurate reservoir budgets are important for understanding regional fluxes of sediment and nutrients. Here we present a comprehensive budget of sediment (based on total suspended solids, TSS), total nitrogen (TN) and total phosphorus (TP) for two subtropical reservoirs on rivers with highly intermittent flow regimes. The budget is completed from July 1997 to June 2011 on Somerset and Wivenhoe reservoirs in southeast Queensland, Australia, using a combination of monitoring data and catchment model predictions. A major flood in January 2011 accounted for more than 50% of the water entering and leaving both reservoirs in that year, and more than 30% of water delivered to and released from Wivenhoe over the 14 year study period. The flood accounted for an even larger proportion of total TSS and nutrient loads: in Wivenhoe 40% of TSS inputs and 90% of TSS outputs between 1997 and 2011 occurred during January 2011. During non-flood years, mean historical concentrations provided reasonable estimates of TSS and nutrient loads leaving the reservoirs. Calculating loads from historical mean TSS and TP concentrations during January 2011, however, would have substantially underestimated outputs over the entire study period, by a factor of up to ten. The results have important implications for sediment and nutrient budgets in catchments with highly episodic flow. Firstly, quantifying inputs and outputs during major floods is essential for producing reliable long-term budgets. Secondly, sediment and nutrient budgets are dynamic, not static. Characterizing uncertainty and variability is therefore just as important for meaningful reservoir budgets as accurate quantification of loads.





## 1 Introduction

Over the past century, human activities have caused unprecedented changes in water, sediment and nutrient movement between the atmosphere, lithosphere, hydrosphere and biosphere (Rockström et al., 2009). Modification of these natural biogeochemical cycles on a range of scales has the potential to alter fundamental earth system processes and

undermine the ecosystem services on which human societies depend (Steffen et al., 2015; Vörösmarty and Sahagian, 2000). For example, artificial fixation of atmospheric nitrogen by humans exceeds fixation rates by all natural processes combined, contributing to a range of environmental problems including acidification, eutrophication and climate change (Gruber and Galloway, 2008; de Vries et al., 2013). The rate of application of P to erodible soil is unsustainable in many parts of the world (Carpenter and Bennett, 2011) and may threaten future food security (Cordell et al., 2009; Van Vuuren

et al., 2010).

Managing soil and nutrient resources more sustainably is therefore imperative, requiring reliable, quantitative sediment and nutrient budgets at local, regional and global scales (e.g. Syvitski et al., 2005; Radach and Pätsch, 2007; Metson et al., 2012). Reservoirs have a major impact on nutrient and sediment budgets due to their high residence times and burial rates relative to free-flowing rivers (Sherman et al., 2001; Friedl and Wüest, 2002; Bosch and Allan, 2008; Kunz

et al., 2011). Reservoirs are also more effective than lakes at retaining both phosphorus (P) and nitrogen (N) (Harrison et al., 2009; Kõiv et al., 2011). Globally, reservpors are estimated to trap 26 % of the modern export of sediment to the coastal zone, and billions of tonnes of sediment have been impounded within reservoirs since the mid-20th century (Syvitski et al., 2005).

While quantifying sediment and nutrient loads is essential for closing local and regional nutrient budgets (Metson

et al., 2012; Walling and Collins, 2008), estimating uncertainty in these loads is a major challenge (Walling and Collins, 2008; Parsons, 2011; Carpenter et al., 2015). Sediment and nutrient retention in reservoirs depends on many factors, including delivery (which is related to catchment size, land use and geology and river discharge volumes), sediment particle size, storage capacity and water release practices (Issa et al., 2015; Mahmood, 1987; Graf et al., 2010; Leigh et al., 2010). In tropical and subtropical river systems, large and episodic fluctuations in discharge due to seasonal and inter-decadal cycles

in rainfall patterns mean that large sediment and nutrient inputs can be delivered in relatively short time-frames (Kennard et al., 2010; Lewis et al., 2013). For example, in one reservoir in subtropical Australia, net phosphorus retention over a 6-year drought period was driven by moderate-flow events over just 12 days (Burford et al., 2012). Thus reservoir budgets can vary across different time periods (Parsons, 2011). The greater the climatic variability in the catchment, the longer the budget timeframe required to capture representative data.





This study aims to complete budgets of sediment and nutrients (N and P) for two large subtropical reservoirs. The catchments of both reservoirs are characterised by high intensity episodic rainfall and runoff events, therefore the budgets are conducted over more than a decade to capture a wide range of climatic conditions. More specifically, the study assesses the effect of variability in flow on both the magnitude and uncertainty in sediment and nutrient loads entering,

leaving and retained within the reservoirs.

## 2 Materials and Methods

Sediment and nutrient budgets were completed for Somerset and Wivenhoe reservoirs over 14 years from July 1997 to June 2011. For this study, sediment is defined as the mixture of inorganic and organic matter, measured by dry weight of filtered solids, i.e. total suspended solids (TSS). Inputs and outputs of water, TSS, total N (TN) and total P (TP)

were estimated using a combination of catchment model predictions and monitoring data, measured at intervals ranging from hourly to monthly. Output loads of TSS, TN and TP were estimated using four different methods to deal with missing data.

### 2.1 Study area

Somerset and Wivenhoe reservoirs are major drinking water and flood mitigation reservoirs in southeast

Queensland, Australia (27$^{o}$24'S, 152 $^{o}$36' E and 27° 7' S, 152° 33' E, respectively) linked by the Stanley River. The Stanley River was dammed to form Somerset reservoir in 1959, and the Wivenhoe dam wall was constructed further downstream below the confluence of the Stanley River and upper Brisbane River (UBR) in 1984 (Figure 1). The catchment areas of Somerset and Wivenhoe reservoirs are 1 340 and 7 020 km$^{2}$, respectively. At full supply capacity, Somerset holds 0.380 km$^{3}$, with a mean water depth of 9.3 m and a surface area of 42 km$^{2}$. Wivenhoe holds 1.165 km$^{3}$ with a mean water depth

of 10.5 m, and surface area 107 km$^{2}$ (Leigh et al., 2015). Both reservoirs are eutrophic and warm-monomictic, with overturn in the austral autumn and stratification in the austral summer that results in anoxic bottom waters (Burford and O'Donohue, 2006). Water is released continuously from Wivenhoe reservoir for water treatment downstream.

Mean annual rainfall in the region is 743 mm (Bureau of Meteorology, bom.com.au, Figure 2). Inflows enter Somerset reservoir primarily from the Stanley River. Controlled releases from Somerset reservoir combine with inflows

from the Upper Brisbane River (UBR) and lateral inflows to supply Wivenhoe reservoir (Figure 1). Stanley River and UBR have highly unpredictable and intermittent flow regimes (Kennard et al., 2010), although major discharge events tend to occur in summer (the 'wet season'). Therefore, 'water years' were defined from July to June in all analyses to capture the entire austral summer wet season within each water year.





During the study period, there were three above-average flow events of note: in February 1999 (water year 1998), February 2008 (water year 2007) and January 2011 (water year 2010). From 9-16 January 2011, a large flood with extreme rainfall occurred within the Wivenhoe and Somerset catchments (Seqwater, 2011). It was the second highest flood recorded in the lower Brisbane River over the past century (the highest was in 1974), and water from the Wivenhoe
catchment contributed to significant flood damage downstream (van den Honert and McAneney, 2011). The February 1999 and 2008 events were small by comparison (e.g. as indicated by rainfall volumes in Figure 2; see also van den Honert and McAneney, 2011). Therefore, water year 2010 (July 2010-June 2011) is referred to hereafter as the "flood year" and all other water years during the study period are denoted as "non-flood years". The non-flood years (July 1997 to June 2010) comprised a range of hydrological conditions, including the 1999 and 2008 flow events and the 2001-2009 drought (Dijk et
al., 2013) which was characterised by low rainfall and low inflows to both reservoirs (Leigh et al., 2015).

### 2.2 Catchment inputs: flows, loads and uncertainty

Daily flow and TSS, TN and TP loads from the catchments into Somerset and Wivenhoe reservoirs were estimated using the SourceCatchments (SC) model (Weber et al., 2009). The model was parameterized for hydrology and stream routing using one stream gauge in the Somerset catchment, and four stream gauges in the Wivenhoe catchment. The
15 model used global, land-use-based event-mean concentrations (EMC) and dry weather concentrations (DWC), estimated from water quality information collected across the southeast Queensland region, with a particular focus on those event monitoring sites that adequately characterized the pollutant export from land uses and soil types typical of the regions being modelled. Calibration and validation were undertaken using a combination of manual and automated techniques.

Uncertainty in the SC model was estimated by comparing SC predictions with flow and loads measured at two
gauging stations: Woodford Weir on the Stanley River, and Gregors Creek on the UBR (Figure 1, Table 1). Flow volume recorded for the gauging stations spanned many orders of magnitude, making it difficult to distinguish between zero flow and missing data. Therefore, model predictions were only compared with non-zero recorded flows. When plotted against flow measured at the gauging stations, SC predictions in both the Stanley River and UBR were scattered around the 1:1 line on the log-log scale (Figure S1). Variability between gauged and predicted water input was highest at low flow, and lowest
when predictions and data were integrated to a yearly time-step (Figure S1). The adjusted $R^2$ for log annual flow (SC) vs log annual flow (gauged) was 0.96 for the Stanley River and 0.95 for the Upper Brisbane River (UBR), and the 95 % confidence interval for the slope contained 1.0 for both rivers. Root mean square error of the difference between measured and predicted flow was 70% of the mean annual flow, when averaged across the Stanley and UBR gauging stations for available data during the study period.



Uncertainty in SC predictions of TSS and nutrients was more difficult to quantify, due to the limited data availability (Table 1). TN, TP and TSS loads predicted by SC were compared with event loads measured at the Stanley River and UBR gauging stations (Figure 1), for 32 and 15 high flow events, respectively, during the study period (Table 1). During each high flow event, the concentrations of TSS, TN and TP were measured by the local water authority, Seqwater, at the gauging stations. Water samples were automatically collected using a refrigerated autosampler triggered by the change in height of the depth-gauge above a base-flow threshold. Loads were determined using the linear-interpolation method, with at least 10 measurements per event, and sampling on both rising and falling limbs of the hydrographs (Olley et al., 2014). TSS, TN and TP loads predicted by SC were well correlated with the loads estimated from the event-sampling data collected for both gauging stations (Figure S2). Total loads across all measured events differed from SC predictions by 24% for TSS, 45% for TN and 26% for TP (and 42% for flow) when averaged across the two sites, verifying that the SC predictions were consistent with flow and loads in the major tributaries during high flow events. However this information did not provide a measure of uncertainty in annual predicted loads to the two reservoirs.

To estimate uncertainty in annual inputs, an empirical model was used to predict TN and TP loads at each gauging station based on measured daily flow (Kerr, 2009; Burford et al., 2012). The model was validated against the event loads (Figure S2) and then compared with SC predictions at daily, monthly and annual time steps using gauged flow data (Figure S1). Unfortunately an empirical model was not available for TSS.

During high flow events, the empirical model predictions agreed with the SC predictions and the measured TN and TP event loads (Figure S2). Daily, monthly and annual predictions of both TN and TP from the empirical model agreed with SC predictions (Figure S1). Difference between the two models was lowest for the Stanley River site (Woodford Weir), even though the empirical model was developed for the UBR. Variation between the models was lowest when information was integrated to an annual time-step (Figure S1). Over the entire study period, the root mean square difference between the two models as a proportion of mean annual loads was 60% for TN and 45% for TP, when averaged across the Stanley and UBR gauging stations. Uncertainty could not be estimated for TSS, and flow was the only variable for which SC predictions could be directly compared with data. Uncertainty in loads is unlikely to be lower than uncertainty in flow, which was estimated as 70 %, as outlined above. Therefore we assumed an uncertainty of 70% in annual SC model predictions of flow, TSS, TN and TP inputs to both reservoirs (Table S1).

### 2.3 Reservoir outputs

Loads of TSS, TN and TP exported from the reservoirs each month were calculated by multiplying concentrations ([TSS], [TN] and [TP], in mg L$^{-1}$) measured at the dam walls by the volumes of water released. The volume of monthly water





released from each reservoir was determined by summing daily release values, except during the period 1 July 1997 to 30 June 2001 for Wivenhoe reservoir, for which monthly release data were directly available (Table 1).

### 2.3.1 Data sources: [TSS], [TN] and [TP] at dam walls

Concentrations of TSS, TN and TP in water released from the reservoirs were determined from routine monthly

monitoring and sub-daily turbidity profiles collected near the dam walls.

Monthly monitoring data collected by Seqwater were available for surface and bottom concentrations of TSS, TN, TP, ammonium ($NH_4$), nitrite plus nitrate ($NO_2+NO_3$), dissolved inorganic P (DIP) at the dam wall of each reservoir from July 1997 to June 2011. Surface samples were taken using a 3 m depth-integrated sampler and bottom samples were taken using a van Dorn sampler. Samples were filtered as needed in situ, stored on ice until frozen in the laboratory and

processed using standard methods (Burford and O'Donohue, 2006; APHA, 1995).

Depth profiles of turbidity (NTU) were also measured at the dam wall in each reservoir, recorded by a calibrated nephelometer deployed on a fixed buoy. Turbidity profiles at 1 m intervals through the water column were available approximately every hour for water years 2009-2010 in Somerset reservoir and water years 2008-2010 in Wivenhoe reservoir (Table 1).

In Somerset reservoir, water release occurs when the dam gates open from the bottom. For low release volumes, "bottom" waters are released, but at higher release rates water from higher in the water column will be entrained. To account for this, the concentrations of nutrients and TSS in the Somerset release water were assumed equal to bottom concentrations when daily release was < 500 ML d$^{-1}$. At higher flows (i.e. ≥ 500 ML d$^{-1}$), TSS and nutrient concentrations in the water released were assumed equal to the average of surface and bottom concentrations. Wivenhoe reservoir is a

near-surface water-releasing reservoir, so monthly exports of nutrients and TSS were calculated from surface concentrations only.

### 2.3.2 Estimating sediment and nutrient loads from turbidity profiles

Monthly monitoring data were available for the entire 14 year study period, but data were missing for December 2010 and January 2011, when release volumes and turbidity were both unusually high (Grinham et al., 2012). Turbidity

profiles were available for December 2010 and January 2011, but were only available for a short portion of the entire study period (two water years in Somerset and three water years in Wivenhoe, Table 1). Hence the datasets needed to be combined in some way to provide a meaningful long-term budget for the reservoirs.





The turbidity profile data could only be used to estimate loads released from the reservoirs if a meaningful relationship could be established between turbidity and [TSS], [TP], and [TN]. It is quite common to develop local relationships between [TSS] and turbidity. Since P is strongly associated with sediment particles, a relationship between turbidity and [TP] might also be expected. However dissolved compounds typically make up a large component of [TN], so

a strong relationship between turbidity and TN was not be expected.

Routine monthly surface and bottom measurements of [TSS], [TN] and [TP] were correlated with mean daily surface and bottom turbidity measured on the same day, where data from both sources were available (Table 1). Daily surface and bottom turbidity were determined from readings in the top 3 m and the bottom 2 m respectively, averaged across each day. Since the objective was to determine concentrations during turbid floodwaters when routine monitoring

was unavailable, [TN] and [TP] were only used where NTU > 15. Turbidity data were cleaned prior to analysis: spikes associated with calibration were removed by inspection. Where gaps in the record were no greater than two days, they were replaced with the average turbidity of the preceding and subsequent day.

Linear regression in matlab was used to determine the correlation coefficients for the relationship described by:

$$[y] = a + b \text{ NTU} \tag{1}$$

where [y] is [TSS], [TN] or [TP], and $a$ and $b$ are the corresponding intercept and slope (Table S2). Eq (1) was then used to calculate daily estimates for [TSS], [TN], [TP] from surface and bottom mean daily turbidity.

### 2. 3.3 Reservoir outputs calculated from multiple data sources

There were a number of ways in which the monthly monitoring and turbidity profile data could be combined to calculate sediment and nutrient outputs from Somerset and Wivenhoe reservoirs over the study period. We compared four

such methods of estimating output loads:

- *Method 1 Mean historical concentration*: Surface and bottom [TSS], [TN] and [TP] at dam wall sites in each reservoir were estimated from the mean concentration of monthly monitoring data 1997-2011 (Table S3). This had the advantage of a consistent data source for the full timeframe of the study, and was justified because variation in release volume is orders of magnitude above variation in [TSS], [TN] and

[TP] at the dam wall. However this method may underestimate the output loads of TSS, TN and TP during very large floods, when water leaving the reservoir has unusually high TSS and nutrient concentrations (e.g., Lewis et al., 2013). Note that mean [TSS] was determined from log-transformed data, due to small numbers of very high values;



- *Method 2 Monthly measured concentration*, with missing data replaced by mean historical concentration (as defined in Method 1). This makes better use of the information available, but will not provide much advantage over Method 1 in dealing with the flood year, since monitoring data were unavailable for December 2010 and January 2011, when large volumes of water were released and turbidity at the dam wall was very high (Grinham et al., 2012);

- *Method 3 Monthly measured concentration*, with missing data replaced by information from turbidity profiles where available, and by mean historical concentration otherwise. This enables better estimation of [TSS], [TN] and [TP] during January 2011, and does not rely on turbidity correlations where direct measurements of those concentrations are available;

- *Method 4 Concentration calculated from turbidity profiles*, with missing data replaced by monthly measured concentrations where available, and by mean historical concentration otherwise. This makes best use of the high resolution turbidity profile information, but relies strongly on the correlation between turbidity and [TSS], [TN] and [TP].

The output loads of TSS, TN and TP used in the final budget were calculated from Method 3. The uncertainty in budget output loads (Table S1) was estimated at 40 % of TSS and TP and 10 % for TN, based on the relative mean difference between annual loads predicted by Methods 3 and 4 for the only non-flood years for which turbidity data were fully available: Somerset water year 2009, and Wivenhoe water year 2008-2009 (Table S1).

**2.4 Reservoir budgets: inter-annual comparisons and propagation of error**

Annual accumulation of TSS, TN and TP in each reservoir was calculated as the sum of catchment inputs (SC model predications) and loads from the upstream reservoir (in the case of Wivenhoe), minus reservoir outputs. Where data were combined (e.g. Wivenhoe input loads were the sum of SC model predictions and Somerset output loads), uncertainty was determined using the law of propagation of errors, assuming that errors were independent (Ku, 1966). Thus errors in total loads over a given timeframe ($\Delta \sum_{i=1}^{n} X_i$) were calculated from the square root of the sum of squares of errors in individual loads ($\Delta X_i$):

$$\Delta \left( \sum_{i=1}^{n} X_i \right) = \sqrt{\left( \sum_{i=1}^{n} \Delta X_i^{\ 2} \right)} \qquad (2)$$



Relative error in mean load was assumed to equal relative error in total load. Annual retention of TSS, TN and TP for each reservoir was compared against hydraulic retention time (reservoir volume at full supply divided by annual inflow volume). Trapping efficiency (*TE*) was calculated from input and output loads as follows:

$$TE = \frac{Input - Output}{Input} \qquad (3)$$

In accordance with the law of propagation of errors, again assuming errors in input and output loads are independent (Ku, 1966), the uncertainty in trapping efficiency *ΔTE* was calculated from the relative errors in input and output loads (*ΔInput/Input* and *ΔOutput/Output* respectively) as follows:

$$\Delta TE = (1 - TE)\sqrt{\left(\frac{\Delta Input}{Input}\right)^2 + \left(\frac{\Delta Output}{Output}\right)^2} \qquad (4)$$

**3 Results**

The flood year (water year 2010: July 2010-June 2011) dominated inputs and outputs of water, sediment and nutrients for both reservoirs. Inputs of water, TSS, TN and TP to Somerset and Wivenhoe were 5-10 times higher in 2010 than on average during the 13 non-flood years (Figure 3, Table 2). Reservoir outputs were approximately 10-50 times higher than during the non-flood years (Figure 3, Table 2). The biggest effect of the flood year was on output of TSS, which was 40 - 50 times higher in the flood year. Wivenhoe inflows were particularly impacted: whereas the input of water, sediment and nutrient to both reservoirs was very similar during non-flood years, inputs to Wivenhoe were more than double those to Somerset during the flood year.

The flood month, January 2011, also had a major impact on the reservoir budgets. The volumes of water entering and leaving Somerset and Wivenhoe during January 2011 (i.e. 0.6% of the study period) accounted for 50% and 60% respectively of the loads for the 2010 water year, and 10% and 30% respectively of the loads over the entire study period (Table 2, Figure 4). The impact of the flood month on the total budget was greatest for TSS and nutrient loads. Based on [TSS], [TN] and [TP] estimated from the turbidity profiler at the dam walls, the loads of TSS and nutrient outputs from Somerset during January 2011 accounted for 50 -70% of output loads during water year 2010, and 20-50% of output loads over the study period (Figure 4). The flood month had the greatest impact on Wivenhoe: TSS and nutrient exported in January 2011 accounted for 70-90% of export loads during the water year, and 40-70% of export loads over the entire 14 year study period.



Inter-annual variability in water-release volumes from both reservoirs was much higher than variability in the [TSS], [TN] and [TP] at the dam wall during non-flood years (Figures 3, S3), implying that variation in reservoir output was driven by variation in the volume of water released rather than the concentrations of sediments and nutrients in the water. As a result, there was little difference between output loads estimated from historical mean concentrations (Method 1)

and from monthly monitoring (Method 2) during non-flood years (Figure 5). The only non-flood year for which turbidity data was available for both reservoirs was 2009, and there was little difference between loads calculated using mean concentrations, monthly monitoring data or [TSS], [TN] and [TP] calculated from the turbidity profiler at the dam wall for that year (Methods 1-4, Figure 5).

The combination of extremely high releases and unusually high turbidity, however, meant long-term historical

mean concentrations did not provide a good estimate of reservoir outputs of TSS or TP during the flood year (Figure 5). Monthly monitoring data was unavailable during January 2011 (Table 1), when turbidity, inflows and releases of water were very high for both reservoirs (Figure 6). If TSS and nutrient outputs were estimated from mean concentrations (Methods 1 or 2), the TSS export during January 2011 and water year 2010 would have been underestimated by an order of magnitude (Table S4). Additionally, TP output loads during this period would have been underestimated by a factor of two

in Somerset, and five in Wivenhoe. However the mean concentrations provided a reasonable estimate for TN loads, because TN concentrations were less affected by the flood than TSS or TP (Figure 6).

TSS trapping efficiency was very high during the non-flood period, regardless of the hydraulic residence time (Figure 7). While the majority of TN and TP delivered to both reservoirs over the entire non-flood period was retained (Table 3), Wivenhoe was a net exporter of TN in many water years (Figure 7) due to high concentrations of dissolved

inorganic N accumulating in the bottom waters of the reservoir (Figure S3). In water year 2010 the net retention or export of water, TSS, TN and TP was less than the bounds of uncertainty (Table 3), with the exception of retention of TSS in Somerset.

As noted earlier, both the flood year and flood month had greater effects on Wivenhoe than Somerset. Wivenhoe has three times the full supply volume of Somerset, and four times the catchment area. Despite the difference in

catchment area, mean inputs to Wivenhoe and Somerset were very similar during the non-flood period (hence the hydraulic retention time was typically shorter for Somerset, as shown in Figure 7). However during the flood year, inputs to Wivenhoe were double or triple those to Somerset (Table 2, Figure 3). Wivenhoe receives water from two sources: controlled releases from Somerset and episodic inputs from the catchment, which are dominated by flows from the UBR. Catchment flows account for about half (50-60%) of water inflows and the majority of TSS and nutrient inputs in both flood

and non-flood years (Figure 3).





[TSS], [TN] and [TP] measured in the main tributary supplying inflows to Wivenhoe, the UBR, were typically greater than in water leaving the reservoirs (Figure S3). The proportion of dissolved nutrients and the N:P ratios, however, differed between the reservoirs and the river inputs (Figure S4). DIP concentrations were higher in the UBR than in either of the reservoirs, while dissolved inorganic N (DIN) concentrations where higher in the bottom waters of the reservoirs than in either the UBR or surface waters of the reservoirs (Figure S3). As a result, DIN:DIP and TN:TP ratios and the proportion of TN in readily bioavailable form (DIN) were all higher in the bottom of the reservoirs than in the rivers (Figure S4). In all cases, a higher proportion of P than N was available in dissolved inorganic form, and DIP:TP was higher in the UBR than in the reservoirs.

**4 Discussion**

**4.1 Flood impacts on reservoir budgets: implications for monitoring and management**

Our budget calculations show that the January 2011 flood dominated inputs, outputs and retention of sediment and nutrient for both reservoirs over the 14 year study period. We have very high confidence in this conclusion because the inputs calculated here for January 2011 represent a lower bound estimate. The catchment model and reservoir release data in this study predicted that 2.1 TL of water flowed into Wivenhoe during the peak of the flood (9-16 January 2011), which is 26 % lower than the 2.64 TL inflow estimated by Seqwater (2011). TSS input to Wivenhoe in January 2011 was estimated by Grinham et al. (2012) as 1.8 Mt, based on event mean concentrations, and 21 Mt, based on a correlation between flow and TSS. These estimates are one and two orders of magnitude respectively above our estimate of 0.2 Mt (Table 2). Event mean concentrations do not account for the shape of the flood peak, and there is an order of magnitude difference between the loads estimated from the event mean concentration method and the flow-load correlations. This demonstrates the difficulty not only in determining loads for reservoir budgets, but also in finding meaningful estimates of uncertainty.

Our uncertainty analysis was as thorough as possible given the data available, but our estimate of 70 % confidence in SC model predictions may not be valid for major floods. While the predictions of TSS loads generated by the SC model agreed well with measured loads in flow events at gauging stations on both the Stanley River and UBR, the January 2011 event was so large in magnitude that it was outside the calibration range of the SC model and the rating curves at the gauging stations. Refining the estimates of input and output loads during January 2011 is the key to both reducing and better quantifying uncertainty in long-term sediment and nutrient budgets for the reservoirs.

Reliable reservoir budgets require reliable data. During non-flood years, mean historical concentrations provided reasonable estimates of TSS and nutrient loads leaving the reservoirs. However calculating loads from historical mean TSS and TP concentrations during January 2011 would have underestimated outputs over the entire study period by a factor of





2-10 (Figure 5, Table S4). Since extreme flow events generate both the highest inputs and outputs of TSS and nutrients, and the highest uncertainty in loads, more intensive monitoring data from high flow events is required to increase confidence in these long-term reservoir budgets. Reducing the frequency of routine monitoring and using these savings to fund measurements during extreme events may therefore be a cost-effective way to reduce uncertainty in reservoir budgets.

The hydrological regimes of both Somerset and Wivenhoe are typical of the unpredictable and intermittent flow regimes found in rivers on the eastern coastal fringe of Australia (Kennard et al. 2010). Hence our findings will be particularly relevant in tropical and subtropical systems, where intra- and inter-annual variability are particularly high (Lewis et al. 2013). Because major floods play such a dominant role in the sediment and nutrient budgets of reservoirs with highly variable flow regimes, sustainable management of soil and nutrient resources will mean addressing sediment

erosion and nutrient inputs during major floods. Land use change is the key factor responsible for changes in sediment and nutrient delivery to downstream water bodies throughout Australian catchments and no doubt in similarly modified landscapes beyond (Harris, 2001; Bartley et al., 2012; Powers et al., 2015). In the subtropical catchments of southeast Queensland reservoirs, for example, river channel erosion is the main source of sediment inputs, and restoring riparian vegetation is the main mechanism by which these loads can be reduced (Wallbrink, 2004; Leigh et al., 2013; Olley et al.,

2014).

### 4.2 Uncertainty and variability in reservoir budgets

The three catchment budget principles of Parsons (2011) provide a useful framework for understanding the results of this study. Parsons proposed that 1) specifying the timeframe of validity for catchment budgets is important; 2) quantities determined from the difference between measured loads should be treated with caution; and 3) uncertainty

should be specified on all values. Our results highlight connections between these three principles. For example, the timeframe of the budget affects the uncertainty in budget estimates in two ways. Firstly, if there are no systematic errors in budget loads, relative error in total loads will decline as duration of the study increases, as can be seen from Eq (2). This explains why relative uncertainty in mean loads over the non-flood years and retention over the entire study period are much lower than uncertainty during the flood year (Table 3). Secondly, budgets conducted over longer timeframes are

more likely to capture a realistic representation of climatic conditions, particularly in tropical and subtropical systems where variation in flow can be extremely high (Kennard et al., 2010; Burford et al., 2012; Lewis et al., 2013). Variation in input and output loads was very high even in the 13 non-flood years (Fig 3); the standard deviation of input and output loads was typically similar or equal to the mean load for both reservoirs (Table 2).

While quantifying uncertainty in reservoir budgets is important (Parsons, 2011), it can be extremely difficult, due

to the necessity of combining data and predictions from different sources, across different spatial and temporal scales (Walling and Collins, 2008; Hobgen et al., 2014). In this study, we were able to quantify uncertainty in all loads, using a





range of methods, including verification of the catchment model SC against both event loads and independent empirical models. Relative uncertainty was highest in reservoir retention (Table 3), because retention is the difference between input and output loads, and uncertainty in retention depends on the addition of input and output errors squared (Eq 2). This means the magnitude of retention is less than either input or output loads, but uncertainty in retention is higher than in

either inputs or outputs: hence relative uncertainty in retention can be very large (Tables 2-3). Full quantification of uncertainty in all components of the budget (Parsons' third principle of catchment budgets) makes it clear than uncertainty is particularly high in "unmeasured elements" (Parsons' second principle).

Thus if uncertainty is quantified for all budget terms, and the variability in the system is adequately accounted for, then the three principles of catchment budgets proposed by Parsons (2011) will be met. In systems such as our study sites,

where flow is highly episodic, a static budget of water, sediment or nutrient loads will have limited value, and budgets are best presented as time series.

### 4.3 Sediment and nutrient trapping

Correct propagation of uncertainty also affects interpretation of reservoir budgets. Comparison of the magnitudes and uncertainties in retention and trapping efficiency of water, sediment and nutrients (Tables 2-3) clearly illustrates this

point, as follows. Net retention of TSS, TN and TP occurred over the 14 year study period in both reservoirs, except for TP in Somerset, where uncertainty was higher than the difference between input and output loads. The flood year dominated the retention of TSS, TN and TP in both reservoirs (e.g. 25 % and 40 % of TSS retained in Somerset and Wivenhoe were captured during the flood year). Uncertainty is higher over shorter time periods, however, as outlined earlier, so retention of water, sediment and nutrients in both reservoirs in the flood year was only significantly different to zero for TSS in

Somerset. In contrast, trapping efficiency (retention divided by input) was quantifiable for all sediment and nutrients across the study period, and for TSS in both reservoirs and TN in Somerset during the flood year. Together, these findings engender greater confidence in the proportion of sediment and nutrients retained by the reservoirs (i.e. trapping efficiency) than in the mass retained. For a fuller assessment of trapping efficiency in reservoirs with variable flow, such as Wivenhoe and Somerset, hydraulic retention on shorter timescales is required (Lewis et al., 2013).

Retention of sediments in reservoirs can represent a loss of terrestrial productivity, and reduce the volume available for water supply and flood mitigation. For example, sedimentation in Mosul Dam, Iraq, reduced reservoir volume by more than 10 % between 1986 and 2011 (Issa et al., 2015). To determine volume occupied by sediment retained in Somerset and Wivenhoe over our study period, we divided the mass of sediment retained (Table 3) by an estimated sediment bulk density of 0.95 gcm$^{-3}$, using the appropriate unit conversions. The sediment bulk density used here

represented an average of the range reported by Avnimelech et al. (2001). For Wivenhoe, we used TSS inputs from two sources for January 2011: 1. TSS inputs from this study (Table 2) and 2) mean TSS input estimated by Grinham et al. (2012;



11.4 ± 9.6 Mt). In the most extreme case (i.e. highest estimates of sediment inputs during January 2011), Wivenhoe storage volume is estimated to decline by only 1 % over the 14 year study period (Figure 5). Using the input loads calculated in this study, decline in storage volume over 14 years is estimated as only 0.04-0.1 % (Table 4). Direct measurement of reservoir volume is required for more accurate estimates of storage loss due to sedimentation.

Clear differences between TSS, TN and TP retention were observed across both reservoirs, reflecting the different processing pathways of sediment, nitrogen and phosphorus in aquatic systems. TSS trapping was very high, with lower variability and relative uncertainty than TN and TP, and a stronger correlation to hydraulic residence time (Figure 7). This reflects sediment dynamics, which are strongly controlled by the physical processes of advection and settling. TP retention was lower and more variable than TSS retention in either reservoir, but was also related to hydraulic residence time (Figure

7), similar to the findings of a long-term study of an arid lake system in Australia (Cook et al., 2010). P retention has been demonstrated in reservoirs throughout the world (Josette et al., 1999; Bosch and Allan, 2008). However TP retention was more variable than TSS retention because P can be transformed via chemical and biological processes into a range of organic and inorganic forms. TP is associated with the finer fractions of TSS, which are less likely to settle and hence more likely to be transported through the reservoir during periods of short retention time (Kerr et al., 2011), increasing the

proportion of P likely to be transported through the reservoir during periods of overflow. A full analysis of residence time impact on TSS and TP retention is likely to require flow data on shorter timescales, e.g. daily rather than annual inflow (Lewis et al., 2013).

     Interpreting retention of N is more complicated than either TSS or TP. Whereas both nutrients and sediments can be deposited from the atmosphere and buried in sediments, N can also be exported via denitrification and imported

through N fixation by cyanobacteria. These processes are not included in the budget, thus uncertainty in TN loads and retention will be underestimated. N is typically retained in reservoirs globally (Harrison et al., 2009), and was consistently retained in Somerset throughout the study period. However Wivenhoe was frequently a net exporter of TN (Figure 7), typically during drought years when Wivenhoe releases for Brisbane water supply were less than reservoir inflows (Figure 3).

The impact of reservoirs on downstream aquatic ecosystems depends of the form of nutrients released as well as the total loads (Kunz et al., 2011). Overall, TN is retained by both reservoirs over the study period (Table 3). However the concentration of dissolved inorganic nitrogen (DIN) leaving the bottom of both reservoirs was typically higher than the concentration of DIN measured in the UBR during events (Figure S3), probably due to anoxic conditions in reservoir bottom waters (Burford and O'Donohue, 2006). Ratios of total and dissolve inorganic N: P were substantially higher in the

reservoirs compared with the UBR. Therefore the impacts of reservoirs on downstream nutrient conditions will depend on



the timing and magnitude of sediment and nutrients loads into the reservoirs, trapping efficiency and transformation processes within the reservoirs themselves.

## 5. Conclusions

Major floods are likely to dominate long term sediment and nutrient budgets in reservoirs such as Somerset and Wivenhoe subject to episodic flow. Accurate quantification of inputs and outputs during high flow periods is therefore essential for reliable sediment and nutrient budgets. For these two subtropical reservoirs a static budget of water, sediment or nutrients is meaningless at best, and misleading at worst, because both the magnitude and timing of loads are highly dynamic. Understanding variability and uncertainty are therefore just as important as quantifying loads in characterizing reservoir budgets in regions with intermittent and variable flow. This is especially relevant in a world in which many once-perennial rivers are expected to transition to intermittent flow regimes (Döll and Schmied, 2012) and the pace of dam construction in many regions continues to escalate (Winemiller et al., 2016).

## 6. Acknowledgments and Data

We thank Andrew Watkinson, Kate Smolders and the staff at Seqwater for the long-term reservoir data, Badin Gibbes and Alistair Grinham for useful discussions, Stephen Faggotter for the map and Phil Keymer for data checking. This project was funded by Seqwater and an ARC Linkage project LP0776375 "Sources of phosphorus promoting cyanobacteria in subtropical reservoirs". MAB has a long-term research collaboration with Seqwater. We also thank X reviewers for their comments on the manuscript. Data supporting the conclusions will be submitted to an online data repository once this manuscript is accepted for publication.

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





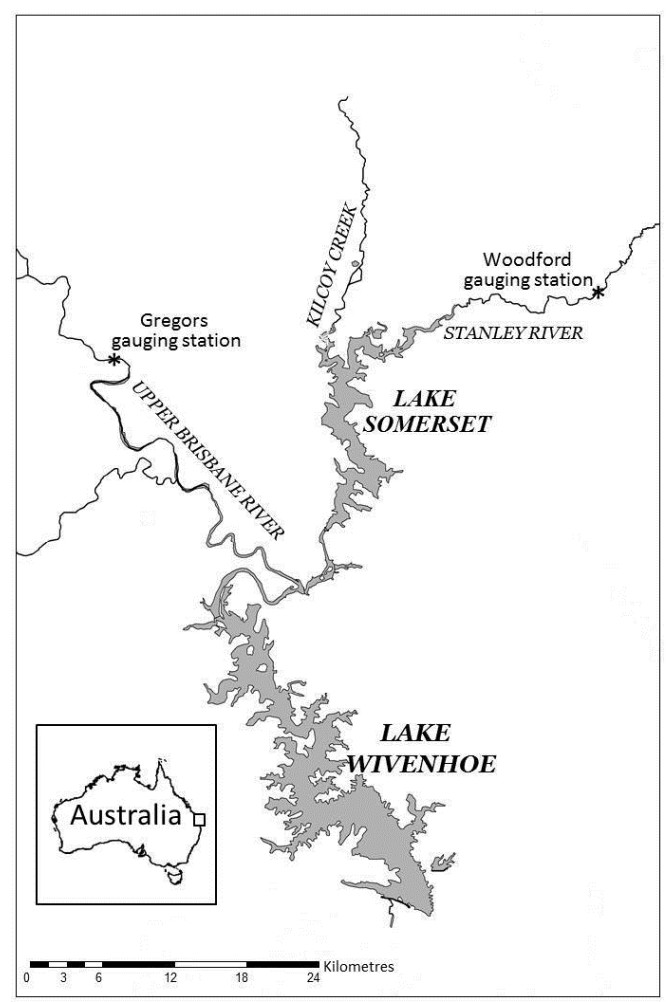

**Figure 1: Somerset and Wivenhoe reservoirs in subtropical Australia. The major tributaries are Stanley River and Upper Brisbane River (UBR), respectively. Flow gauging stations are indicated.**




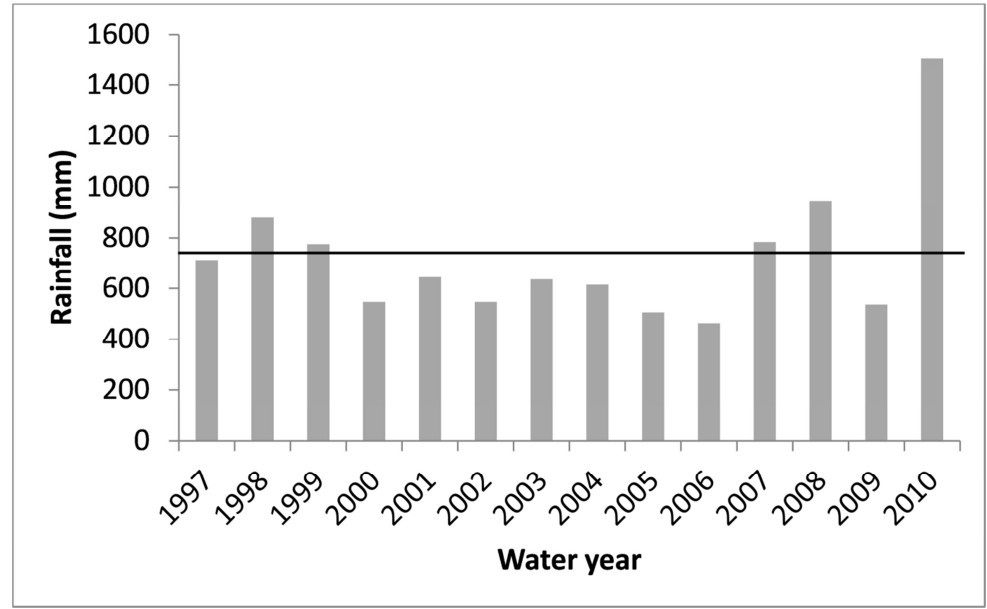

**Figure 2: Annual rainfall (mm per water year) measured at a rainfall station near Wivenhoe and Somerset reservoirs (Bureau of Meteorology, bom.com.au). Horizontal line shows long-term mean rainfall.**





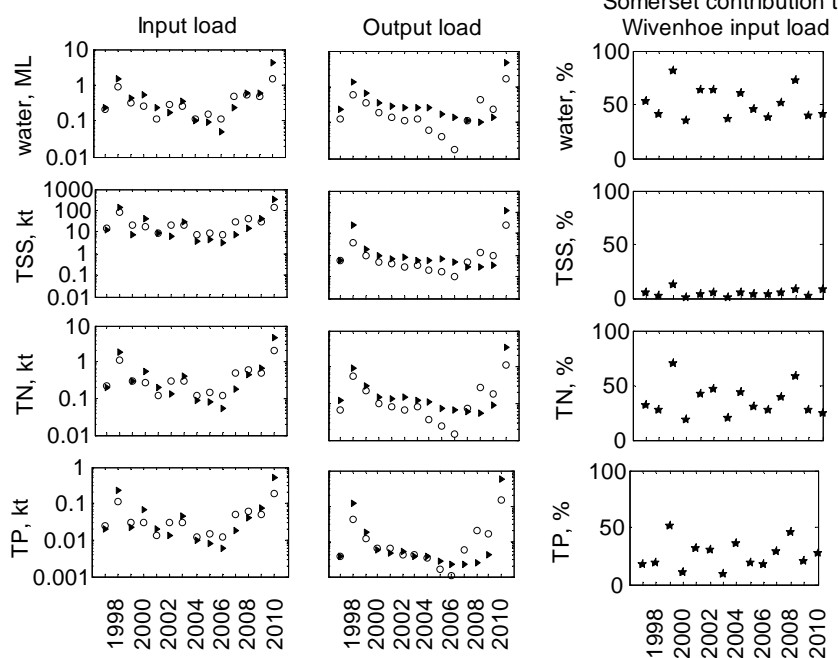

**Figure 3: Annual input and output loads of water (ML y⁻¹), TSS and nutrients (kt y⁻¹) for Somerset (o) and Wivenhoe (►) reservoirs for water years 1997-2010, and the percentage contribution of Somerset to Wivenhoe input loads.**





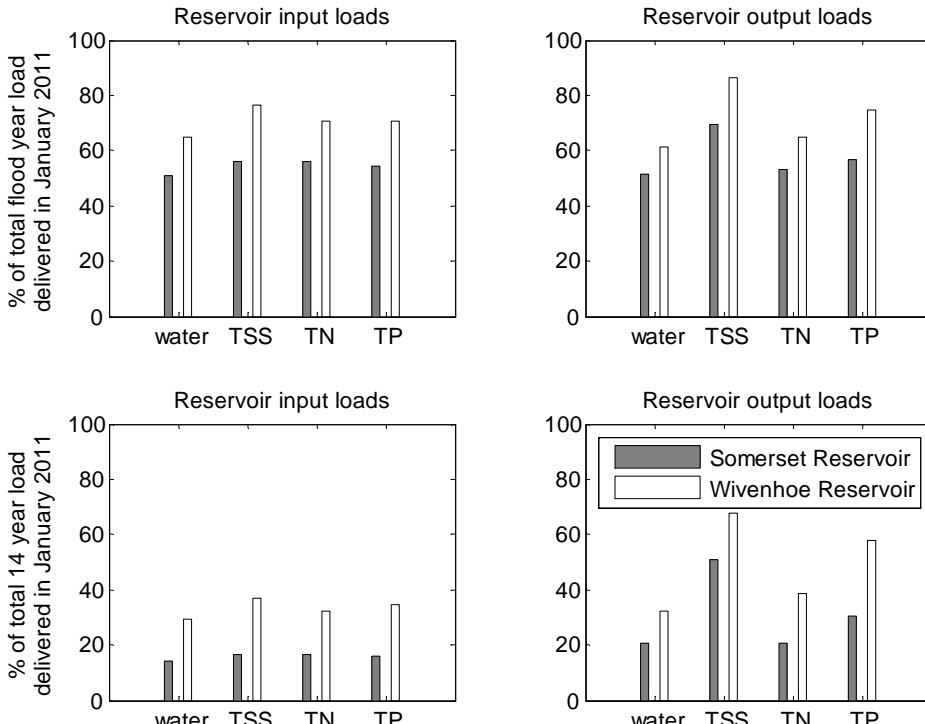

**Figure 4: Summary of January 2011 input and output loads, as percentage of total loads in and out during the flood year, and across the entire study period**





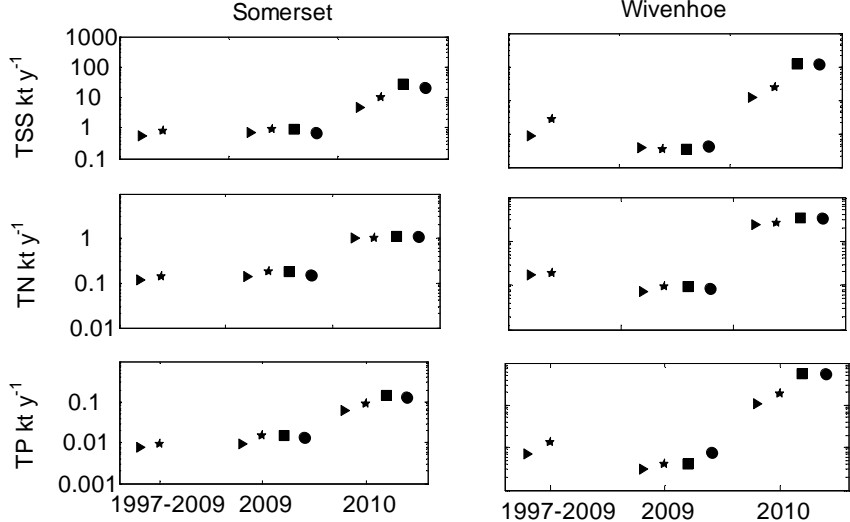

**Figure 5: Comparison of four methods for calculating mean annual TSS, TN and TP output loads (kt y⁻¹), using [TSS], [TN] and [TP] from: 1. ► Mean historical concentration of monthly monitoring data water years 1997-2010; 2. ∗ Monthly monitoring, with missing data replaced by mean historical concentration; 3. ■ Monthly monitoring, with missing data replaced by concentration estimated from turbidity profiles, and mean historical concentration where turbidity data unavailable; 4. ● Turbidity profiles, with missing data replaced by monthly monitoring, and mean historical concentration otherwise.**





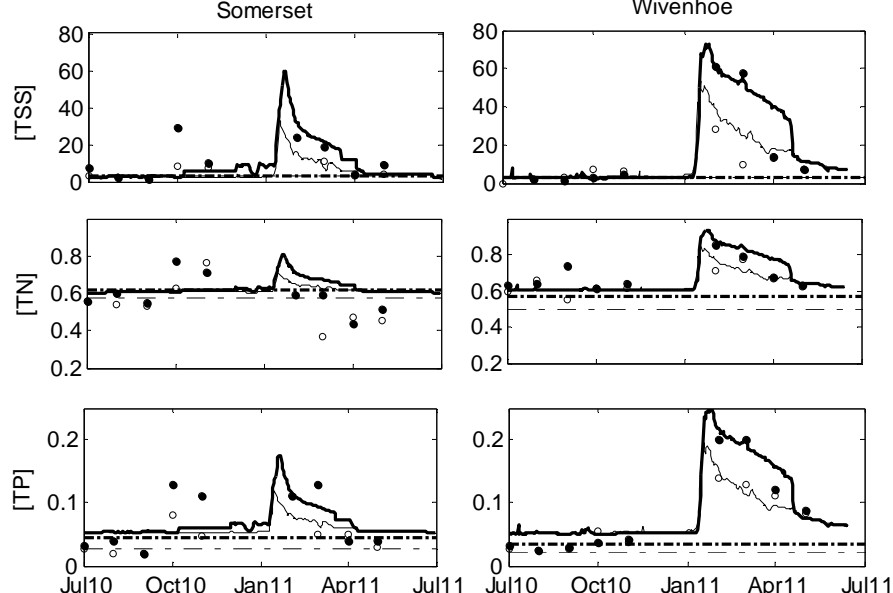

**Figure 6:** [TSS], [TN] and [TP] (mg L⁻¹) at dam outlets measured during monthly monitoring (round symbols), calculated from the daily measured turbidity profile (solid lines) and mean historical concentrations (broken lines). Surface concentrations are denoted by open circles and thin lines, bottom readings are closed circles and heavy lines. Note that TSS mean is from log-transformed data.




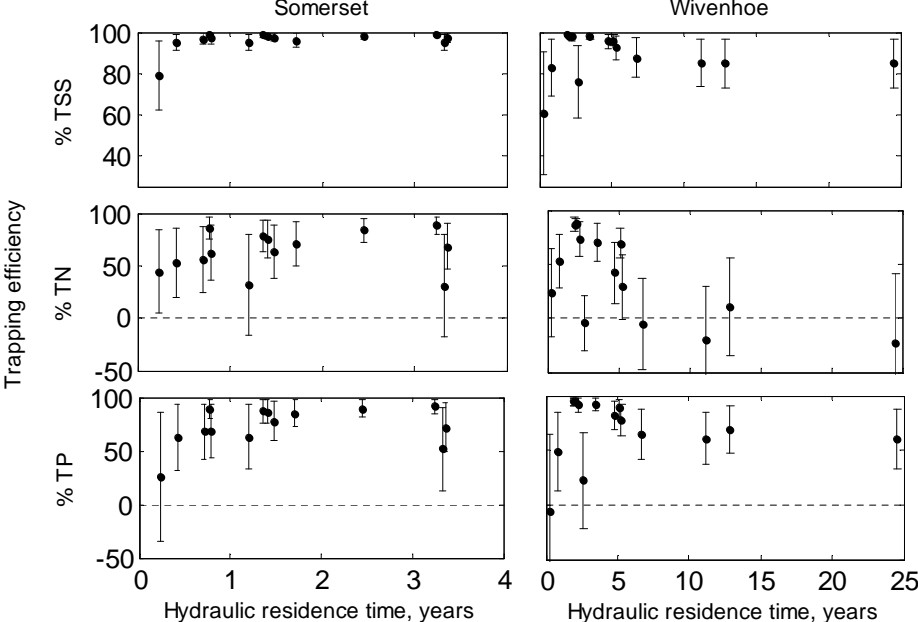

**Figure 7: Percentage of annual TSS, TN and TP loads retained in Somerset and Wivenhoe reservoirs compared to hydraulic residence time (y). Dashed line indicates zero trapping, boundary between net positive import and export.**





**Table 1: Summary of information used to construct reservoir budgets, including spatial and temporal resolution. S = Somerset reservoir, W = Wivenhoe reservoir.**

| | Catchment inputs | Reservoir releases |
|---|---|---|
| **Flow** | *Flow gauged at sub-daily time-step* | *Monthly data* |
| | S: Woodford Weir, Stanley River Jul 2002-Jun 2011 | S & W: Jul 1997 – Jun 2001 |
| | W: Gregors Creek, UBR Jul 1997-Jun 2011 | *Daily data* |
| | *Daily catchment flow predicted by Source Catchments (SC) model* S & W: Jul 1997 – Jun 2011 | S & W: Jul 2001 – Jun 2011 |
| **TSS, TN, TP** | *Daily catchment loads predicted by Source Catchments (SC) model-* S & W: Jul 1997 – Jun 2011 | *Concentrations measured monthly at the dam wall, top and bottom of water column* S & W: Jul 1997 – May 2011 |
| | *Catchment inputs at gauging stations estimated from flow and concentration measured during high flow events* | *Concentrations estimated from turbidity profiles taken hourly throughout the water column at the dam wall* |
| | S: 32 events at Woodford Weir, Stanley River , 6 Dec 2003 -26 Jul 2009 | S: Jul 2009 – Jun 2011 |
| | W: 15 events at Gregors Creek, UBR 25 Dec 2002- 3 Jul 2009 | W: Jul 2008 – Jun 2011 |
| | *TN, TP only: Daily loads at Woodford Weir and Gregors Crossing estimated from daily gauged flow using an empirical model (Kerr 2009)* | |
| | S & W: Jul 2002 – Jun 2011 | |





**Table 2: Inputs and output loads of water, TSS, TN and TP for Somerset and Wivenhoe reservoirs from June 1997 to July 2011 ± uncertainty, σ is the standard deviation of annual values over non-flood years. Water years are defined from July to June. Water year 2010 is the flood year, other years are non-flood years. January 2011 is the flood month. S = Somerset reservoir, W = Wivenhoe reservoir**

| | | Input loads | | | Output loads | | |
|---|---|---|---|---|---|---|---|
| | | *Mean, non-flood years* | *Flood year* | *Flood month* | *Mean, non-flood years* | *Flood year* | *Flood month* |
| **Water, $10^9$ m$^3$** | S | 0.32± 0.07 σ= 0.21 | 1.6 ±1.1 | 0.8 | 0.20 σ= 0.16 | 1.7 | 0.9 |
| **($10^3$ GL)** | W | 0.39 ±0.06 σ= 0.36 | 4.2±1.8 | 2.7 | 0.33 σ= 0.32 | 4.8 | 2.9 |
| **TSS, kt** | S | 23±6 σ= 19 | 130 ±90 | 73 | 0.8±0.1 σ= 0.9 | 27± 10 | 19 |
| | W | 25±8 σ= 37 | 310 ±200 | 235 | 2.6±1 σ= 7 | 120± 50 | 104 |
| **TN, kt** | S | 0.36±0.1 σ= 0.27 | 2.0 ±1.4 | 1.1 | 0.13±0.01 σ= 0.14 | 1.1 ±0.1 | 0.6 |
| | W | 0.4 ±0.1 σ= 0.5 | 4.5 ±2.4 | 3.1 | 0.18 ±0.01 σ= 0.22 | 3.4 ±0.3 | 2.2 |
| **TP, kt** | S | 0.04±0.01 σ= 0.03 | 0.2±0.14 | 0.1 | 0.01±0.002 σ= 0.01 | 0.14±0.06 | 0.1 |
| | W | 0.04±0.01 σ= 0.05 | 0.5±0.3 | 0.4 | 0.01±0.004 σ= 0.03 | 0.57±0.2 | 0.4 |





**Table 3: Retention of water, TSS, TN and TP in Somerset and Wivenhoe reservoirs from June 1997 to July 2011. Water year 2010 is the flood year. S = Somerset reservoir, W = Wivenhoe reservoir**

| | | Retention | | Trapping efficiency = Retention/Input loads | |
|---|---|---|---|---|---|
| | | *Entire study period* | *Flood year* | *Entire study period* | *Flood year* |
| **Water, 10⁹ m³** | **S** | 1.44±1.46 | -0.14 ±1 | 25±19% | -9±76% |
| **(10³ GL)** | **W** | 0.21±1.91 | -0.56 ±2 | 2 ±20% | -13 ±47% |
| **TSS, kt** | **S** | 400± 120 | 100 ± 90 | 92 ±3% | 79 ±60% |
| | **W** | 480± 230 | 190 ±200 | 76 ±12% | 61 ±30% |
| **TN, kt** | **S** | 3.8±1.8 | 0.9 ±1.4 | 57 ±12% | 44 ±39% |
| | **W** | 4.0±2.6 | 1.1 ±2.4 | 41 ±16% | 24 ±41% |
| **TP, kt** | **S** | 0.40±0.19 | 0.05 ±0.1 | 60 ±14% | 26 ±60% |
| | **W** | 0.36±0.39 | -0.03 ±0.36 | 33±28% | -5±69% |





Table 4. TSS retention and estimated decline in storage capacity for Somerset and Wivenhoe reservoirs from June 1997 to July assuming sediment bulk density of 0.95 gcm$^{-3}$. * calculated from information in Table 2; ** input of TSS in water year 2010 based on January 2011 TSS loads estimated by Grinham et al., 2012; all other information from Table 2.

| | TSS retention, kt | Total decrease in storage capacity, km$^3$ | Relative decrease in storage capacity |
|---|---|---|---|
| **Somerset**[*] | 400 ± 120 | 0.0042 ± 0.00013 | 0.11 ± 0.03 % |
| **Wivenhoe**[*] | 480 ± 230 | 0.0051 ± 0.00024 | 0.04 ± 0.02 % |
| **Wivenhoe**[**] | 11600 ± 9600 | 0.012 ± 0.01 | 1.1 ± 0.9 % |