# Peer review of "Sediment and nutrient budgets are inherently dynamic: evidence from a long-term study of two subtropical reservoirs"

_Hydrology and Earth System Sciences, 2016_

## Referee Comment (RC1) · Anonymous Referee #1 · 19 May 2016

General comments:

This manuscript uses a series of different methods to construct input and output streamflow and TSS, TN and TP loads/budgets for two reservoirs situated in sub-tropical southeast Queensland. The results are used to provide an estimate of un-certainty in the loads and are then used to calculate the retention of the constituents. The key finding is that the major flood in 2010/11 dominated the sediment and nutrient budgets for the entire 14 year study period and highlights the importance of long-term monitoring with a primary focus on obtaining high resolution data from large flood pe-riods. The data also show that while the high uncertainty in the load data in the flood period prevents reliable calculations of mass retained in the reservoirs during this time (the uncertainty was higher than the mass retained), the trapping efficiency as a per-centage could still be estimated for some parameters as well as the retention over the

whole study period. Overall the findings provide important insights for research on catchment budgets and reservoir retention and the paper is suitable for publication. I initially found the construction of the manuscript to be a little unconventional where it appeared study results (i.e. modelled and monitored load data and uncertainties, comparisons between the data etc) were reported in the 'materials and methods' section with this section being quite long relative to results/discussion. However, on reflection given the main findings/theme of the manuscript I don't think a restructure would greatly improve the manuscript. Hence I recommend publication pending some minor revisions that relate to methods that require additional information and some minor textual edits.

Specific comments:

Section 2.2: I'm wondering if some more information (or a reference) could be provided for the Source Catchments/WaterCAST modelling from the SEQ region? For example the model captures land uses soil type EMCs, however, later in the manuscript it is stated that river channel erosion is the main source of sediment inputs to the reservoirs – can the model account for this process?

Page 4 Lines 27-28: Some more information of the methods of analysis for TSS and in particular TN and TP would be helpful. Can a reference be provided on the analysis? E.g. What digestion was used; what instrument were the data measured etc. Also might be worth pointing out that there is uncertainty in the laboratory analysis as well!

Page 6 Lines 7-11: I was a bit confused on this paragraph on what data the authors are referring to – on a second read I'm guessing that it was monthly turbidity data (measured on a probe?) that was collected? Then this data was compared with the nephelometer data (the two datasets?). This just needs to be better clarified.

Page 6 Lines 25-27: Not sure if this is needed (what is expected)?

Page 8 Line 21: I may have missed it in the manuscript but I was not clear on what the

difference is in the definition between 'mean load' and 'total load'?

Figure 8: Not sure if this figure adds too much to the manuscript?

Technical comments:

Abstract: Suggest adding brackets for (TN) and (TP) where first defined.

Page 2 Line 13: Please spell out 'nitrogen' so reads '...and sometimes nitrogen (N)...'

Page 2 Line 29: 'This study' = new paragraph so needs tab space. Also note a few instances in the manuscript where the tab for new paragraph is needed – Page 3 Line 19, Page 3 Line 25, Page 4 Line 14, Page 4 Line 24 (starting with Uncertainty), Page 5 Line 8.

Page 3 Line 10: Please add '(SEQ)' after 'southeast Queensland' (then okay to use SEQ later on – e.g. Page 4 Line 11)

General comment – please check spaces throughout the manuscript (e.g. Page 3 Line 15; Page 4 Line 4)

Page 3 Line 26: Suggest adding 'catchment area' so will read '...within the Wivenhoe and Somerset catchment area..'

General comment – 'data are plural' so e.g. Page 5 Line 27 'Monthly monitoring data were' (also check Page 6 Line 7; Page 7 Line 25; Page 9 Line 27; Page 10 Line 5).

Page 5 Line 28: Replace 'was completed by' with 'from'?

Page 6 Line 27: (If keeping this section) delete 'be'.

Page 9 Lines 15-16: Suggest replacing 'loads' in these two lines with 'water volume' (I was initially confused thinking that these were TSS and nutrient loads and the next sentences were repetitive).

General comment: From ∼ Page 9 Line 17, TSS, TN and TP are reported as [TSS], [TN] and [TP] – suggest being consistent throughout paper (unless there is a distinction

between the two ways that I have missed?).

Page 10 Line 11: Suggest deleting 'retention'

Page 10 Line 13: 'TSS was retained in both reservoirs..... in all non-flood years' – already said two sentences before – so maybe delete?

Page 10 Lines 23-24: The way I read this is that 60% of catchment inflows in 2010 water year was from the upper Brisbane arm meaning that 40% is from Somerset discharge – but during the flood period would all the 40% be 'controlled release'?

Page 11 Line 9: Not sure on this one should it be 'TSS inputs... were estimated' or 'TSS input.... was estimated'?

Page 12 Line 23: replace 'than' with 'that' so reads '..clear that uncertainty...'

Page 13 Line 1: do you mean Wivenhoe instead of Somerset here (for TP retention over study period)?

Page 13 Line 7: Suggest to reference Table 2 in here again.

Page 13 Line 15: Suggest to check through the references – seem how they are listed are a little inconsistent throughout (e.g. should it be Avnimelech et al. (2001)?)

Page 13 Line 20: suggest removing 'however' at end of sentence and add 'for' so will read 'Direct measurement of reservoir volume is required for more accurate estimates of change in storage volume'.

Page 14 Line 14: add 'd' so 'dissolved inorganic N'.
* * *

---

## Author Comment (AC1) · 16 Jun 2016

Author response: We thank the reviewer for their input. We agree with the reviewer's general comments, and suggest the following title change to demonstrate the broader applicability of our findings: Sediment and nutrient budgets are inherently dynamic: evidence from a long-term study of two subtropical reservoirs

Specific Reviewer comments: Reviewer comment: Section 2.2: I'm wondering if some more information (or a reference) could be provided for the Source Catchments/WaterCAST modelling from the SEQ region? For example the model captures land uses soil type EMCs, however, later in the manuscript it is stated that river channel erosion is the main source of sediment inputs to the reservoirs – can the model account for this process?

[Figure]

Author response: The following text and additional reference have been added: "The event mean concentrations (EMCs) used within the Source Catchments model were derived from event monitoring and continuous sampling within the catchments of interest (Thomson et al. 2013), and thus implicitly represent the range of sediment and nutrient generation processes present within the catchment. The EMCs are attributed to land uses rather than specific generation processes. This attribution is relatively consistent with the spatial characterisation of sediment generation within the catchment as quite often the generation processes are strongly tied to the land management of particular land activities (unpublished data). For channel erosion, denuded areas within the river reaches which are aligned to land uses such as horticulture and grazing where land clearing activities have been conducted to the channel edge. Further improvements in the model would require better data representing individual processes which currently doesn't exist for many parts of the catchment studied."

Reviewer comment: Page 4 Lines 27-28: Some more information of the methods of analysis for TSS and in particular TN and TP would be helpful. Can a reference be provided on the analysis? E.g. What digestion was used; what instrument were the data measured etc. Also might be worth pointing out that there is uncertainty in the laboratory analysis as well!

Author response: The text on p4 has been modified as follows: Water samples were automatically collected using a refrigerated autosampler triggered by the change in height of the depth gauge above a base flow threshold. For TN and TP, whole samples were kept on ice until frozen in the laboratory. They were later analyzed using the persulfate digestion method and run through an autoanalyzer (Burford and O'Donohue, 2006; APHA, 1995). For TSS samples, a known volume was filtered onto a pre-weighed and combusted glass fibre filter, then dried and reweighed (APHA, 1995).

Author response: Furthermore the text on p6 has been modified as follows: Reviewer comment: Monthly monitoring data collected by Seqwater were available for surface and bottom concentrations of TSS, TN, TP, ammonium (NH4), nitrite plus nitrate (NO2+NO3), dissolved inorganic P (DIP) at the dam wall of each reservoir from July 1997 to June 2011. Surface samples were taken using a 3 m depth-integrated sampler and bottom samples were taken using a van Dorn sampler. TN and TP samples were kept on ice until frozen in the laboratory. They were later analyzed using the persulfate digestion method and run through an autoanalyzer (Burford and O'Donohue, 2006; APHA, 1995). For dissolved nutrients, samples were filtered through 0.45 $\mu$m membrane filters in situ, and kept on ice until frozen in the laboratory. Samples were analyzed using standard colorimetric methods with an autoanalyzer (Burford and O'Donohue, 2006; APHA, 1995). For TSS samples, a known volume was filtered onto a pre-weighed and combusted glass fibre filter, then dried and reweighed (APHA, 1995).

Reviewer comment: Figure 8: Not sure if this figure adds too much to the manuscript? Author response: We agree: Figure 8 has been removed.

Reviewer comment: Technical comments: Abstract: Suggest adding brackets for (TN) and (TP) where first defined. Author response: Agreed

Reviewer comment: Page 2 Line 13: Please spell out 'nitrogen' so reads '...and sometimes nitrogen (N)...' Author response: Agreed

Reviewer comment: Page 2 Line 29: 'This study' = new paragraph so needs tab space. Also note a few instances in the manuscript where the tab for new paragraph is needed – Page 3 Line 19, Page 3 Line 25, Page 4 Line 14, Page 4 Line 24 (starting with Uncertainty), Page 5 Line 8. Author response: Agreed

Reviewer comment: Page 3 Line 10: Please add '(SEQ)' after 'southeast Queensland' (then okay to use SEQ later on – e.g. Page 4 Line 11) Author response: Agreed Reviewer comment: General comment – please check spaces throughout the manuscript (e.g. Page 3 Line 15; Page 4 Line 4) Author response: Agreed Reviewer comment: Page 3 Line 26: Suggest adding 'catchment area' so will read '...within the Wivenhoe and Somerset catchment area..' Agreed Reviewer comment: General comment – 'data

are plural' so e.g. Page 5 Line 27 'Monthly monitoring data were' (also check Page 6 Line 7; Page 7 Line 25; Page 9 Line 27; Page 10 Line 5). Author response: Agreed Reviewer comment: Page 5 Line 28: Replace 'was completed by' with 'from'? Page 6 Line 27: (If keeping this section) delete 'be'. Page 9 Lines 15-16: Suggest replacing 'loads' in these two lines with 'water volume' (I was initially confused thinking that these were TSS and nutrient loads and the next sentences were repetitive). Author response: Agreed

Reviewer comment: General comment: From âĹij Page 9 Line 17, TSS, TN and TP are reported as [TSS], [TN] and [TP] – suggest being consistent throughout paper (unless there is a distinction between the two ways that I have missed?). Author response: TSS, TN and TP refers to total suspended solids, total nitrogen and total phosphorus respectively, whereas [TSS], [TN] and [TP] refer specifically to concentrations of these materials. This is widely accepted notation, but will be clarified from first usage of the square brackets on p9.

Reviewer comment: Page 10 Line 11: Suggest deleting 'retention' Author response: Agreed

Reviewer comment: Page 10 Line 13: 'TSS was retained in both reservoirs..... in all non-flood years' – already said two sentences before – so maybe delete? Agreed

Reviewer comment: Page 10 Lines 23-24: The way I read this is that 60% of catchment inflows in 2010 water year was from the upper Brisbane arm meaning that 40% is from Somerset discharge – but during the flood period would all the 40% be 'controlled release'? Author response: Yes Reviewer comment: Page 11 Line 9: Not sure on this one should it be 'TSS inputs... were estimated' or 'TSS input.... was estimated'? Author response: Agreed

Reviewer comment: Page 12 Line 23: replace 'than' with 'that' so reads '..clear that uncertainty...' Author response: Agreed

Reviewer comment: Page 13 Line 1: do you mean Wivenhoe instead of Somerset here (for TP retention over study period)? Author response: Agreed

Reviewer comment: Page 13 Line 7: Suggest to reference Table 2 in here again. Author response: Agreed

Reviewer comment: Page 13 Line 15: Suggest to check through the references – seem how they are listed are a little inconsistent throughout (e.g. should it be Avnimelech et al. (2001)?) Author response: Agreed

Reviewer comment: Page 13 Line 20: suggest removing 'however' at end of sentence and add 'for' so will read 'Direct measurement of reservoir volume is required for more accurate estimates of change in storage volume'. Author response: The sentence has been modified as follows"Direct measurement of reservoir volume is required for more accurate estimates of storage loss due to sedimentation."

Reviewer comment: Page 14 Line 14: add 'd' so 'dissolved inorganic N'. Author response: Agreed

References: Thomson, B., D. Orr, B. Ferguson, R. Gardiner, R. Turner, and M. S. J. Warne. 2013. South East Queensland Event Monitoring Program: Sediment and nutrient loads for 2011–2012. Department of Science. Information Technology, Innovation and the Arts, Brisbane.

---

## Referee Comment (RC2) · Anonymous Referee #2 · 8 Jul 2016

General comments

This paper deals with some important issues about the challenges that are faced in some catchments and reservoirs where peak flows, despite very rare, have a huge influence on the load budget. On the same time these peak flows are the most difficult to monitor and represent very short time periods which complicates data interpretation.

The paper is generally well written with a fluent and precise language and very few grammatical errors. The paper is well structured and both data and methods are described in sufficient detail and it is an interesting data set with a long time series of both flow and nutrient and sediment concentrations. In that sense I find that the paper does have a sufficient quality and some relevance for the general readership of HESS. However, I find that the weak part of the paper is the fact that I do not see that this

paper makes a substantial contribution to our current knowledge about nutrient budget estimations or nutrient and sediment transport processes. It is fairly well known that peak flows can contribute substantially to transport of nutrients and sediment and that monitoring of these peak flows are difficult because any time averaging (which is mostly done during normal flow periods) introduces a huge uncertainty on the peak flow load estimates. Therefore I suggest that it could be considered if this paper might be more suitable for a targeted engineering journal, for instance with special interest in reservoir and dam dynamics and their impacts on freshwater ecosystems.

There is an excessive use of references to supplementary material. I find it somewhat problematic that such a large part of the paper relies on supplementary material. In my opinion supplementary material should function as a supplement, not as an essential extension of the paper. I therefore suggest that the supplementary material is critically reviewed and condensed.

Generally I think that the Conclusion section is more a Perspectives section. I suggest that the conclusion should be rewritten to sum up the findings rather than discussing perspectives and implications.

Specific comments

P. 7 line 5: You write that you do not expect a good relationship between TN and turbidity. However in the plot (Fig. S2) the relationship looks just as good as for TP and TSS? Could you comment on this?

P 8 line 15: For output loads the uncertainty is estimated as deviation of Method 3 from method 4, but why is method 4 used? and not one of the others?

P 8 line 6: In method 3 why are loads not calculated based on the two monthly measurements, rather than just one? Would two measurements not give a better estimate, simply due to less interpolation and more real data?

P. 11 line 19: Do you mean flow-TSS correlations as conducted by Grinham et a.

(2012). Slightly confused with what is your method and what is done by others.

P. 13 line 3-4: You repeat what you just said above about the size of the relative uncertainty compared to input and output uncertainty.

P 13 line 7: That uncertainty is high in unmeasured elements is quite trivial I think. You could either leave this out or state it differently.

P. 12-13: There is a really large focus on this other study, but I do not see clearly how this study advances our knowledge compared to the Parson (2011) paper?

P. 13 line 13-15: Do you mean comparison between methods 1-4 or comparison between the two different reservoirs? Please clarify.

P. 13 line 20-24. Are you more confident using trapping efficiency during peak flows or in general? I do not find it completely clear why you come to the conclusion about more confidence in trapping efficiency than in retention, since both are a function of inflow? P. 13. Line 24. Do you have any suggestions to how this could be achieved?

P. 13 line 27 – P 14 line 1-4: I suggest that this should be moved to the results section? I miss a comment of the importance/implications. Do you believe in these numbers, given the uncertainty in loads, and what is then concluded? I suppose that loss of storage volume seems not to be an issue in these two reservoirs, despite and overall net retention of sediment and nutrients?

P 14 line 15-17: Is this your conclusion (this is the impression I get) or one by Lewis et al. (2013)? Either rephrase so that this is clear or delete reference.

P. 14 Line 22: I do not understand this sentence. Less is released and this leads to net export? And where is Brisbane water supply located? Are water pumped from lake Wivenhoe to the water supply? Could you rephrase this sentence?

P 1 line 30 p. 15 line 1-2: I suggest that this section is rewritten to be more specific about this particular study. It is a rather general statement but as I understand it is

based on the findings in this study?

P. 15 line 9-10: This should be moved to the discussion session.

Technical corrections

P. 2 Line 16: please correct typing mistake in "reservpors".

Please be consistent in the use of spelling out "concentration" or writing in in brackets (example p. 10 line 2 and 16).

P. 13 line 6: please replace "than" with "that".

P. 14, line 27: You already defined DIN, no need to repeat it.

P 14, line 29: Please replace "dissolve" with "dissolved" and write "N and P" rather than N:P.

Fig. 6. Please include units on y axis rather than in the figure text.

Figure S3. The figure would be easier to read if the plots were bigger relative to the text.

Figure S4. The figure would be easier to read if the plots were bigger relative to the text.

Table 1. I find this table very difficult to read and I suggest that it is restructured or left out as there is a quite comprehensive description of data in the main text.

Table S4: What does "Method 3: Method 1", is it the deviation between the two?

―――――――――――――――

---

## Author Comment (AC2) · 29 Jul 2016

The following includes the complete review by reviewer 2, and our response to each point and the overall review.
Reviewer 2 General comments: This paper deals with some important issues about the challenges that are faced in some catchments and reservoirs where peak flows, despite very rare, have a huge influence on the load budget. On the same time these peak flows are the most difficult to monitor and represent very short time periods which

complicates data interpretation.

The paper is generally well written with a fluent and precise language and very few grammatical errors. The paper is well structured and both data and methods are described in sufficient detail and it is an interesting data set with a long time series of both flow and nutrient and sediment concentrations. In that sense I find that the paper does have a sufficient quality and some relevance for the general readership of HESS. However, I find that the weak part of the paper is the fact that I do not see that this paper makes a substantial contribution to our current knowledge about nutrient budget estimations or nutrient and sediment transport processes. It is fairly well known that peak flows can contribute substantially to transport of nutrients and sediment and that monitoring of these peak flows are difficult because any time averaging (which is mostly done during normal flow periods) introduces a huge uncertainty on the peak flow load estimates. Therefore I suggest that it could be considered if this paper might be more suitable for a targeted engineering journal, for instance with special interest in reservoir and dam dynamics and their impacts on freshwater ecosystems.

Author response: We thank the reviewer for their detailed response, and constructive advice. The reviewer recognizes the quality and importance of the paper, and noted that it was "well-structured", with "interesting data set with a long time series of both flow and nutrient and sediment concentrations" and of "sufficient quality and some relevance for the general readership of HESS." We agree that the paper is suitable for HESS readership, and that we must ensure it makes a "substantial contribution to our current knowledge about nutrient budget estimations or nutrient and sediment transport processes".

To this end, our paper highlights the hazards of presenting sediment and nutrient budgets in static form, particularly in catchments with highly episodic flow. This finding applies to catchment budgets generally, and is not restricted to reservoirs. The evidence is taken from a comprehensive long-term dataset. Our paper does not therefore simply convey that peak flows contribute to nutrient and sediment budgets or that these

events are difficult to monitor, but goes far further, making a substantial contribution to current knowledge on the fundamental nature of budget estimations and transport processes.

We agree there is room to articulate this important contribution more clearly and have clarified this in our revision, for example by: • Modifying the title to "Sediment and nutrient budgets are inherently dynamic: evidence from a long-term study of two subtropical reservoirs". • Modifying the Discussion, as outlined below, to emphasize the paper's key conclusion: that catchment budgets are inherently dynamic, particularly in river systems with episodic flow. • The Discussion now also explains the significance of the reservoir siltation rates for regional water supply.

Reviewer 2: There is an excessive use of references to supplementary material. I find it somewhat problematic that such a large part of the paper relies on supplementary material. In my opinion supplementary material should function as a supplement, not as an essential extension of the paper. I therefore suggest that the supplementary material is critically reviewed and condensed.

Author response: Constructing a comprehensive catchment budget is difficult, due to the issues in reconciling data collected over a variety of spatial and temporal scales, and in estimating uncertainty (Walling and Collins 2008, Parsons 2011, Carpenter et al. 2015).

We have included in Supplementary Material the calculations and assumptions we used in our robust quantification of uncertainty. This ensures confidence in our final uncertainty estimates, and provides a thorough method for others constructing a catchment budget. This highly detailed and specific technical information would distract from the main findings if included in the main body. Additionally, we are confident that readers who choose not consult this Supplementary Material will understand our paper. We have therefore decided to retain the information in the Supplementary Material. We would certainly agree, under the Editor's advice, to this material being critically

reviewed.

Reviewer 2: Generally I think that the Conclusion section is more a Perspectives section. I suggest that the conclusion should be rewritten to sum up the findings rather than discussing perspectives and implications.

Author response: We differ from the reviewer on this point: we have deliberately focused on the significance of our results, rather than writing a summary. We leave it to Editor to adjudicate this matter of style.

Specific comments Reviewer 2: P. 7 line 5: You write that you do not expect a good relationship between TN and turbidity. However in the plot (Fig. S2) the relationship looks just as good as for TP and TSS? Could you comment on this?

Author response: Relationships with turbidity are commonly used to estimate [TSS], and less commonly for [TP], which is strongly associated with sediment. It is unusual to estimate [TN] from turbidity because dissolved compounds typically make up a large component of total nitrogen, we have done so here because no other data is available during the critical January 2011 flood period. We have modified the text to clarify this point.

Reviewer 2: P 8 line 15: For output loads the uncertainty is estimated as deviation of Method 3 from method 4, but why is method 4 used? and not one of the others?

Author response: We have added the following sentence to the text: "Thus the estimated uncertainty is the difference between loads estimated from monthly monitoring, and the loads estimated from daily turbidity readings. Monthly monitoring and turbidity datasets were both complete for these time periods (water years 2008 and 2009)."

Reviewer 2: P 8 line 6: In method 3 why are loads not calculated based on the two monthly measurements, rather than just one? Would two measurements not give a better estimate, simply due to less interpolation and more real data?

Author response: Monthly monitoring occurred once per month, at the surface and the

bottom. Surface and/or bottom concentrations were used to calculated loads, depending on the method of reservoir release, as explained in section 2.3.1.

Reviewer 2: P. 11 line 19: Do you mean flow-TSS correlations as conducted by Grinham et a. (2012). Slightly confused with what is your method and what is done by others.

Author response: Agreed, this is ambiguous, we have modified the sentence to read "Thus the TSS inputs to Wivenhoe calculated by Grinham et al. (2012) using the event-mean and flow correlation methods are one and two orders of magnitude, respectively, above our estimate of 0.2 Mt (Table 2)."

Reviewer 2: P. 13 line 3-4: You repeat what you just said above about the size of the relative uncertainty compared to input and output uncertainty.

Author response: Agreed, we have deleted the second sentence.

Reviewer 2: P 13 line 7: That uncertainty is high in unmeasured elements is quite trivial I think. You could either leave this out or state it differently.

Author response: Agreed, we have rephrased this to state "uncertainty is particularly high in quantities which are calculated from other budget terms, rather than independently determined".

Reviewer 2: P. 12-13: There is a really large focus on this other study, but I do not see clearly how this study advances our knowledge compared to the Parson (2011) paper?

Author response: We have clarified this significantly in the text, and particularly in the final paragraph of this section: "Therefore we propose that Parsons' three principles of catchment budgets can be refined to two principles: 1. Budgets should be presented as time-series rather than static quantities to clearly display temporal variability and 2. Uncertainty should be quantified for all budget terms, and accounted for in any interpretation of results. "

Reviewer 2: P. 13 line 13-15: Do you mean comparison between methods 1-4 or comparison be- tween the two different reservoirs? Please clarify.

Author response: We agree, this sentence was ambiguous, and has been rephrased as follows "This point is illustrated by the uncertainty in retention and trapping efficiency of water, sediment and nutrients (Tables 2-3), as follows."

Reviewer 2: P. 13 line 20-24. Are you more confident using trapping efficiency during peak flows or in general? I do not find it completely clear why you come to the conclusion about more confidence in trapping efficiency than in retention, since both are a function of inflow?

Author response: Both. We've modified the paragraph as follows to make these points clearer:

"Correct propagation of uncertainty also affects interpretation of reservoir budgets. Uncertainty is higher over shorter time periods, and thus confidence in budget values is lower for the flood year than for the whole study period (Tables 2-3). Net retention of TSS, TN and TP occurred over the 14 year study period in both reservoirs, except for TP in Wivenhoe, where uncertainty was higher than the difference between input and output loads. The flood year dominated the retention of TSS, TN and TP in both reservoirs (e.g. 25 % and 40 % of TSS retained in Somerset and Wivenhoe were captured during the flood year), however the higher relative uncertainty in the values determined for this shorter timeframe means that retention of water, sediment and nutrients in both reservoirs in the flood year was only significantly different to zero for TSS in Somerset.

Uncertainty in trapping efficiency (retention divided by input) is lower than uncertainty in retention, as outlined in Section 2.4. Thus while retention was not significant for most loads during the flood period, trapping efficiency was quantifiable for all sediment and nutrients across the study period, and for TSS in both reservoirs and TN in Somerset during the flood year (Table 2). Together, these findings engender greater confidence in the proportion of sediment and nutrients retained by the reservoirs (i.e. trapping

efficiency) than in the mass retained, and in budget terms calculated for multi-year periods." Reviewer 2: P. 13. Line 24. Do you have any suggestions to how this could be achieved? Author response: The sentence has been modified to clarify this point: "For a fuller assessment of trapping efficiency in reservoirs with variable flow, such as Wivenhoe and Somerset, hydraulic retention should be calculated on shorter (i.e. monthly) timescales, as outlined in Lewis et al. (2013).

Reviewer 2: P. 13 line 27 – P 14 line 1-4: I suggest that this should be moved to the results section?

Author response: Ideally this would appear in the results section, however we feel that it would confuse readers if presented earlier, because the calculations use information from Grinham et al. 2012 which is first introduced in the Discussion. The paragraph also uses the results to draw further conclusions. Therefore we propose to leave this section in the Discussion.

Reviewer 2: P. 13 line 27 – P 14 line 1-4: I miss a comment of the importance/implications. Do you believe in these numbers, given the uncertainty in loads, and what is then concluded? I suppose that loss of storage volume seems not to be an issue in these two reservoirs, despite and overall net retention of sediment and nutrients?

Author response: We have added extra text and an additional reference to verify the numbers, and explain the significance for regional water supply. The additional text reads as follows:

"Using the input loads calculated in this study, decline in storage volume is estimated as only 0.04 %-1.1 % for Wivenhoe over the 14 year study period (Table 4), i.e. 0.003 %-0.1 % per year. Average annual decline in storage volume is two orders of magnitude lower in Wivenhoe compared to Mosul Dam, Iraq, where reservoir volume reduced by more than 10 % due to siltation between 1986 and 2011, i.e. 0.4 % per year on average (Issa et al., 2015). While trapping efficiency of Wivenhoe is slightly less than that

estimated for Mosul Dam, the large difference in siltation between these two reservoirs is due primarily to the difference in sediment loads. Mosul Dam has approximately ten times the storage volume of Wivenhoe, but sediment loads entering Mosul Dam are of order 100-1000 higher than those entering Wivenhoe (Issa et al., 2015).

While the relative siltation rates in both Somerset and Wivenhoe may seem low (Table 4), the corresponding loss in water supply volume is regionally significant. We estimated that the decline in storage capacity over the study period was approximately 4 000 ML for Somerset loss and 5 000- 12 000 ML for Wivenhoe (Table 4). Four of the 15 water supply reservoirs in the region have capacity of less than 5 000 ML, and fewer than half have a capacity greater than 12 000 ML (Leigh et al., 2010). Hence the volume of storage capacity lost in Somerset and Wivenhoe over the 14 year study period is equivalent to the closure of one of more of the smaller reservoirs. Somerset and Wivenhoe supply water to southeast Queensland, a region of rapid population growth which has recently experienced major drought, and where alternatives water sources have much higher greenhouse gas intensity than water supplied from existing reservoirs (e.g. Hall et al. 2011). Therefore any economic assessment of methods to reduce the catchment sediment load in this region should account for costs associated with reservoir siltation and associated loss of water supply volume. Direct measurement of reservoir volume is required for more accurate estimates of storage loss due to siltation."

Reviewer 2: P 14 line 15-17: Is this your conclusion (this is the impression I get) or one by Lewis et al. (2013)? Either rephrase so that this is clear or delete reference.

Author response: This sentence has been deleted.

Reviewer 2: P. 14 Line 22: I do not understand this sentence. Less is released and this leads to net export? And where is Brisbane water supply located? Are water pumped from lake Wivenhoe to the water supply? Could you rephrase this sentence?

Author response: We've clarified the sentence to reduce ambiguity as follows: "However Wivenhoe was frequently a net exporter of TN (Figure 7), typically during drought years when releases for water supply were less than reservoir inflows (Figure 3)."

Reviewer 2: P 1 line 30 p. 15 line 1-2: I suggest that this section is rewritten to be more specific about this particular study. It is a rather general statement but as I understand it is based on the findings in this study?

Author response: Yes, and it has been modified to make this clearer: "Ratios of total and dissolved inorganic N: P were substantially higher in both reservoirs than in the UBR."

Reviewer 2: P. 15 line 9-10: This should be moved to the discussion session.

Author response: As outlined earlier, we differ with the reviewer's opinion on this matter, and await the Editor's decision.

Technical corrections Reviewer 2: P. 2 Line 16: please correct typing mistake in "reservpors".

Author response: corrected.

Reviewer 2: Please be consistent in the use of spelling out "concentration" or writing in in brackets (example p. 10 line 2 and 16).

Author response: As a matter of style, we feel it's preferable to write out concentration in some contexts, and use the bracket notation in others. We will take the Editor's advice on this matter.

Reviewer 2: P. 13 line 6: please replace "than" with "that".

Author response: corrected.

Reviewer 2: P. 14, line 27: You already defined DIN, no need to repeat it.

Author response: DIN replaced with [DIN].

Reviewer 2: P 14, line 29: Please replace "dissolve" with "dissolved" and write "N and

P" rather than N:P.

Author response: corrected.

Reviewer 2: Fig. 6. Please include units on y axis rather than in the figure text.

Author response: corrected

Reviewer 2: Figure S3. The figure would be easier to read if the plots were bigger relative to the text.

Author response: corrected

Reviewer 2: Figure S4. The figure would be easier to read if the plots were bigger relative to the text.

Author response: corrected

Reviewer 2: Table 1. I find this table very difficult to read and I suggest that it is restructured or left out as there is a quite comprehensive description of data in the main text.

Author response: We leave this to the editor's discretion

Reviewer 2: Table S4: What does "Method 3: Method 1", is it the deviation between the two?

Author response: the caption has modified to clarify the difference between the two methods for estimating output loads of sediment and nutrients: "Method 1 uses historical mean concentrations and Method 3 uses monthly monitoring data supplemented by turbidity profile data."

References used in response to reviewer 2: Carpenter, S. R., E. G. Booth, C. J. Kucharik, and R. C. Lathrop. 2015. Extreme daily loads: role in annual phosphorus input to a north temperate lake. Aquatic Sciences 77:71-79. Hall, M. R., West, J., Sherman, B., Lane, J., and de Haas, D.: Long-term trends and opportunities for managing regional water supply and wastewater greenhouse gas emissions, Environmental science & technology, 45, 5434-5440, 2011. Issa, I. E., Al-Ansari, N., Knutsson, S., and Sherwany, G.: Monitoring and evaluating the sedimentation process in Mosul Dam Reservoir using trap efficiency approaches, Engineering, 7, 190-202, doi: 10.4236/eng.2015.74015, 2015. Leigh, C., Burford, M. A., Connolly, R. M., Olley, J. M., Saeck, E., Sheldon, F., Smart, J. C., and Bunn, S. E.: Science to support management of receiving waters in an event-driven ecosystem: from land to river to sea, Water, 5, 780-797, doi:10.3390/w5020780, 2013. Parsons, A. J. 2011. How useful are catchment sediment budgets? Progress in Physical Geography:1-12. Walling, D., and A. Collins. 2008. The catchment sediment budget as a management tool. environmental science & policy 11:136-143.

---

## Author Comment (AC3) · 29 Jul 2016

I have uploaded the revised manuscript, showing all the responses to both reviewer comments in track changes.

A small note: when I first submitted the manuscript, the Editor requested some changes, which I made. It seems that reviewer 1 had access to the first version of the manuscript, while reviewer 2 reviewed the updated version in which I had incorporated the Editor's suggestions.

The manuscript uploaded here is a modified version of my second submission, and deals with all issues raised by both reviewers.

Please also note the supplement to this comment:

[Figure]

http://www.hydrol-earth-syst-sci-discuss.net/hess-2016-89/hess-2016-89-AC3-supplement.pdf

**Supplement:**

* * *

[revised manuscript text omitted]

---

## Author Comment (AC4) · 29 Jul 2016

Supporting Information for

[revised manuscript text omitted]

---

## Referee Report (RR1)

Review of *Sediment and nutrient budgets are inherently dynamic: evidence from a long-term study of two subtropical reservoirs* by Katherine R. O'Brien, Tony R. Weber, Catherine Leigh and Michele A. Burford

**General comments**

The revised paper by O'Brien et al. has been substantially improved. Especially more focus has been directed towards elaborating and underlining the importance of the study, beyond the importance of major flood events in estimating sediment budgets. I also find the changed title more descriptive. Therefore I find that the paper has increased its relevance for the general readership of HESS.

Generally I find that the authors argue well in their reply, and only leave a few points not clarified. Therefore I have three remaining comments that the authors could consider:

*Regarding the supplementary material*: I acknowledge that there is a substantial dataset as well as modelling work behind this paper which has also, as you state, led to the substantial supplementary material that you have included. Therefore I fully accept that for this particular paper you do not intend to condense the supplementary material. However, I maintain my critic that the amount of supplementary material is really on the edge of what is acceptable. At least I did not manage to read the paper without consulting the supplementary material. For instance I would much rather have had figure s3 or table s4 in the paper instead of table 1. In the guideline for manuscript writing to HESS it is stated (Point 2 under guidelines for supplement): "The supplement **shall contain only complementary information** but no scientific interpretations or findings/messages that would go beyond the contents of the manuscript." I think this goes for most journals and therefore I think that you should critically consider your use of supplementary material in future papers. If it gets too extensive and evolves to tell its own "story" it might be a better idea to split the study into two shorter and more concise papers. It is much easier for the reader which is ultimately the one you want understands your findings

*Table 1*: There is no reply from the authors to the comment made about this table. For future papers I give the advice that the authors explain their reasons for either following or not following the reviewer's recommendations rather than referring to the editor. This will save time, both for the editor and the reviewer. Since you have not argued why you prefer to keep the table as it is, I would like to repeat the comment: I find that table 1 is rather confusing. There is a lot of information and it is written in entire sentences which makes it hard to read. I think you give a quite comprehensive description of data in the text which makes the table redundant. I advise the authors to restructure the table using headings and cues rather the sentences or simply leave out the table, since the information is repeated in the text. Or explain why they think that this table extends the reader's understanding.

*Manuscript conclusions*: About the last two sentences in the Conclusion section the authors reply: "As outlined earlier, we differ from the reviewer's opinion on this matter, and await the Editor's decision". As far as I can see there is not an earlier statement about disagreement and

you also changed the conclusion section to be more true the common understanding of a conclusion: that is should conclude on the findings and main implications of the study and not elaborate on the findings. Therefore I am not sure what you mean with your reply, so just a few general words about conclusions that you might find useful for future work: Many journals, including HESS, do not require a conclusion, and you can either simply stop with the discussion or write a small summary if you do not like the repetition required in a conclusion. However, if you do choose to include a conclusion you should conclude in it and not use it as a perspectives section.

Generally I think that the conclusion is so widely used in scientific papers because it is actually a really good help for the reader to get an overview and recapture the main findings of often very complex studies. I agree that conclusions can be rather tedious for the authors to write but remember that they are very useful for the readers.

**Technical corrections:**

Page 4, line 26: Please use "does not" instead of "doesn't".
Page 7 line 18: Please delete "be" after "was not".
Page 15 line 12: I believe that "alternatives" should be singular?
Page 15 line 16: Please delete the last "." in the sentence.